# Hippocampus ghrelin signaling mediates appetite through lateral hypothalamic orexin pathways

Ted M Hsu[1,2], Joel D Hahn[3], Vaibhav R Konanur[1], Emily E Noble[1], Andrea N Suarez[1], Jessica Thai[1], Emily M Nakamoto[2], Scott E Kanoski[1,2]*

[1]Human and Evolutionary Biology Section, Department of Biological Sciences, University of Southern California, Los Angeles, United States; [2]Neuroscience Program, University of Southern California, Los Angeles, United States; [3]Neurobiology Section, Department of Biological Sciences, University of Southern California, Los Angeles, United States

**Abstract** Feeding behavior rarely occurs in direct response to metabolic deficit, yet the overwhelming majority of research on the biology of food intake control has focused on basic metabolic and homeostatic neurobiological substrates. Most animals, including humans, have habitual feeding patterns in which meals are consumed based on learned and/or environmental factors. Here we illuminate a novel neural system regulating higher-order aspects of feeding through which the gut-derived hormone ghrelin communicates with ventral hippocampus (vHP) neurons to stimulate meal-entrained conditioned appetite. Additional results show that the lateral hypothalamus (LHA) is a critical downstream substrate for vHP ghrelin-mediated hyperphagia and that vHP ghrelin activated neurons communicate directly with neurons in the LHA that express the neuropeptide, orexin. Furthermore, activation of downstream orexin-1 receptors is required for vHP ghrelin-mediated hyperphagia. These findings reveal novel neurobiological circuitry regulating appetite through which ghrelin signaling in hippocampal neurons engages LHA orexin signaling.

*For correspondence: kanoski@usc.edu

**Competing interests:** The authors declare that no competing interests exist.

## Introduction

Feeding behavior rarely occurs as a direct result of an immediate metabolic or nutrient deficiency. Rather, most animals (including humans) have habitual feeding patterns, consuming a fixed number of meals at approximately the same time each day. Environmental stimuli associated with learned meal anticipation elicit preparatory biological 'cephalic' responses (e.g., pancreatic insulin release) that occur prior to food consumption (*Woods and Ramsay, 2000*; *Woods et al., 1970*). These types of conditioned biological responses are advantageous when environmental factors place constraints on eating (e.g., predator threat, limited periods of food availability, and social or professional obligations) because they promote consumption of a large quantity of food within a relatively short time period. One important biological signal associated with conditioned appetite and feeding is ghrelin, a hormone derived from P/D1 cells in the stomach that is the only known circulating hormone that increases food intake(*Tschöp et al., 2000*; *Cowley et al., 2003*). While often referred to as a 'hunger' hormone, ghrelin is perhaps more aptly described as a meal-anticipatory hormone based on findings from both humans and rodents showing that ghrelin levels rise before an anticipated feeding bout independent of energy deprivation levels (*Frecka and Mattes, 2008*; *Cummings et al., 2001*; *Drazen et al., 2006*). Genetic deletion of the ghrelin receptor (growth hormone secretagogue receptor 1A; GHSR) eliminates food anticipatory activity under meal entrainment feeding schedules

**eLife digest** Eating can occur for many different reasons. We often eat not simply because we are hungry, but because we have learned to eat at specific times of day or in response to factors in our environments (for example, an advert that reminds of us of something tasty). Despite this, few researchers have investigated how biological signals from the gut communicate with the brain to enable this type of learned, or "conditioned", feeding behavior.

Brain cells called neurons communicate with each other in specific circuits that can cross between different brain regions. Hsu et al. have now developed a "disconnection neuropharmacology" approach, which alters whether the neurons in different brain regions can communicate with each other, to investigate the effects signaling molecules have on brain activity and behavior. Using this approach in rats revealed a new circuit in the brain that controls learned feeding behavior through ghrelin, a hormone that is released from the stomach and increases appetite.

Hsu et al. found that the hippocampus, a brain region important for learning and memory control, uses ghrelin as a signal to engage in learned feeding behavior. Neurons in the hippocampus that respond to ghrelin communicate with other neurons in a region of the brain called the lateral hypothalamus (known for its role in feeding) that produce a signaling molecule called orexin. This circuit therefore links memory and feeding behavior.

Future work will investigate the neural circuits between the hippocampus and other brain regions that are important for feeding behavior, and the hormones that are important for triggering activity in these circuits.

(*Blum et al., 2009*; *Davis et al., 2011*), further supporting the notion that ghrelin acts as a conditioned meal-anticipatory signal.

The mechanisms through which ghrelin communicates with the central nervous system to control learned aspects of food intake regulation are poorly understood. The hippocampus is a brain region historically associated with visuospatial and episodic memory function that has more recently been linked with the control of various feeding-relevant behaviors (*Benoit et al., 2010*; *Davidson et al., 2007*; *Kanoski, 2012*; *Parent et al., 2014*), including conditioned meal anticipation (*Poulin and Timofeeva, 2008*; *Inglis et al., 1994*; *Henderson et al., 2013*). GHSR is expressed abundantly in hippocampal neurons (*Mani et al., 2014*), and we recently demonstrated that pharmacological activation of GHSR by ghrelin in ventral/temporal subregions of the hippocampus (vHP) potently stimulates food intake by increasing both meal frequency and meal size (*Kanoski et al., 2013*). However, the extent to which endogenous vHP GHSR signaling controls conditioned anticipatory feeding behavior has not been investigated.

The lateral hypothalamic area (LHA) is implicated in the control of fundamental motivated behaviors, including food intake (For review see: [*Stanley et al., 2011*; *Leininger, 2011*; *Berthoud and Münzberg, 2011*]). Classic studies show that LHA electrical ablation results in profound hypophagia (*Teitelbaum and Epstein, 1962*; *Anand and Brobeck, 1951*), whereas LHA electrical stimulation robustly increases food intake (*Delgado and Anand, 1953*). The LHA has direct connections with brain regions involved in the regulation of energy balance, including the hypothalamic arcuate and paraventricular nuclei (*Hahn and Swanson, 2015*), and the medial nucleus of the solitary tract (*Ciriello et al., 2013*). In addition, the LHA is also directly connected with several brain regions that regulate 'higher-order' cognitive aspects of feeding behavior, including the vHP, amygdala, nucleus accumbens (ACB), and ventral tegmental area (VTA) (*Hahn and Swanson, 2015*; *Hahn and Swanson, 2010*; *Cenquizca and Swanson, 2006*; *Hahn and Swanson, 2012*; *Goto et al., 2005*). Of note, vHP pyramidal neurons in field CA1 and the subiculum are a major source of direct and exclusively ipsilateral input to the LHA, especially the dorsal perifornical LHA (dpLHA) (*Hahn and Swanson, 2015*; *Cenquizca and Swanson, 2006*; *Hahn and Swanson, 2012*; *Kishi et al., 2000*). The dpLHA also has abundant expression of the neuropeptide orexin (ORX/hypocretin) (*Swanson et al., 2005*; *Hahn, 2010*), which in addition to an established role in behavioral arousal (*Hagan et al., 1999*; *Chemelli et al., 1999*), also has a central orexigenic role in the control of ingestive behavior (*Willie et al., 2001*; *Rodgers et al., 2002*; *Sakurai, 2014*; *Thorpe et al., 2005*; *Sweet et al., 2004*; *Olszewski et al., 2009*). We hypothesize that LHA ORX neurons are a critical downstream substrate

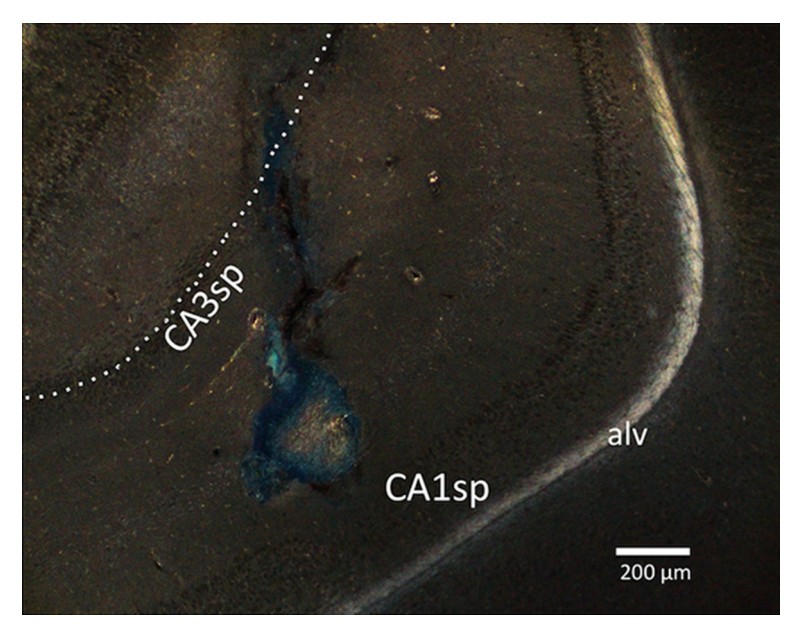

**Figure 1.** Representative vHP injection site is shown as localization of pontamine sky blue ink following a 100 nl injection. CA1sp, CA3sp: CA1 and CA3 pyramidal cells; alv = alveus.

of vHP ghrelinergic appetitive neural responding, because [1] orexin receptor 1 (Ox1R) signaling is required for ventricular ghrelin-induced hyperphagia (*Cone et al., 2014*), and [2] LHA ORX neurons are activated in anticipation of feeding under meal entrainment schedules (*Jiménez et al., 2013*).

In the present study we investigate the neural substrates through which ghrelin communicates conditioned appetite and feeding behavior to and within the brain. We hypothesize the existence of a neural pathway whereby peripherally-derived ghrelin (secreted in anticipation of feeding) activates GHSR-expressing vHP glutamatergic (pyramidal) neurons, which in turn activate orexin-expressing dpLHA neurons to stimulate appetite and feeding via CNS Ox1R signaling. We employed a combination of methods to facilitate a correlative approach, including neural pathway-tracing, immunohistochemistry, behavioral neuropharmacology, meal entrainment conditioning, and excitotoxic unilateral lesion-mediated neural 'disconnection' between pharmacological vHP GHSR activation and the dpLHA. The results of these experiments suggest novel mechanisms linking mnemonic processes with gut-brain communication to control feeding behavior.

## Results

### Endogenous ghrelin signaling in vHP neurons promotes conditioned feeding behavior

*Figure 1* depicts a representative 100nl vHP injection site. Rats restricted to 4 hr chow access per day (meal entrainment schedule) learned to gradually increase their chow consumption over the 8 day training period (significant main effect of Day, p<0.01) (*Figure 2A*). Bilateral parenchymal delivery of the selective GHSR antagonist, JMV2959 (10 µg) to vHP neurons significantly reduced 4 hr chow intake in meal-entrained rats (p<0.05; *Figure 2B*). However, the same dose of JMV2959 had no effect on 4 hr rebound feeding following 20 hr food restriction in control rats that were not meal entrained (*Figure 2C*). Moreover, the JMV2959-induced intake reduction observed in meal-entrained rats appears to be specific to conditioned feeding behavior as opposed to being based on higher baseline kcal consumption in meal entrained vs. control rats. This is supported by results showing that vHP JMV2959 administration did not reduce 4 hr Western diet intake in non-meal entrained, non-food restricted rats under conditions in which total kcal intake exceeded levels observed in

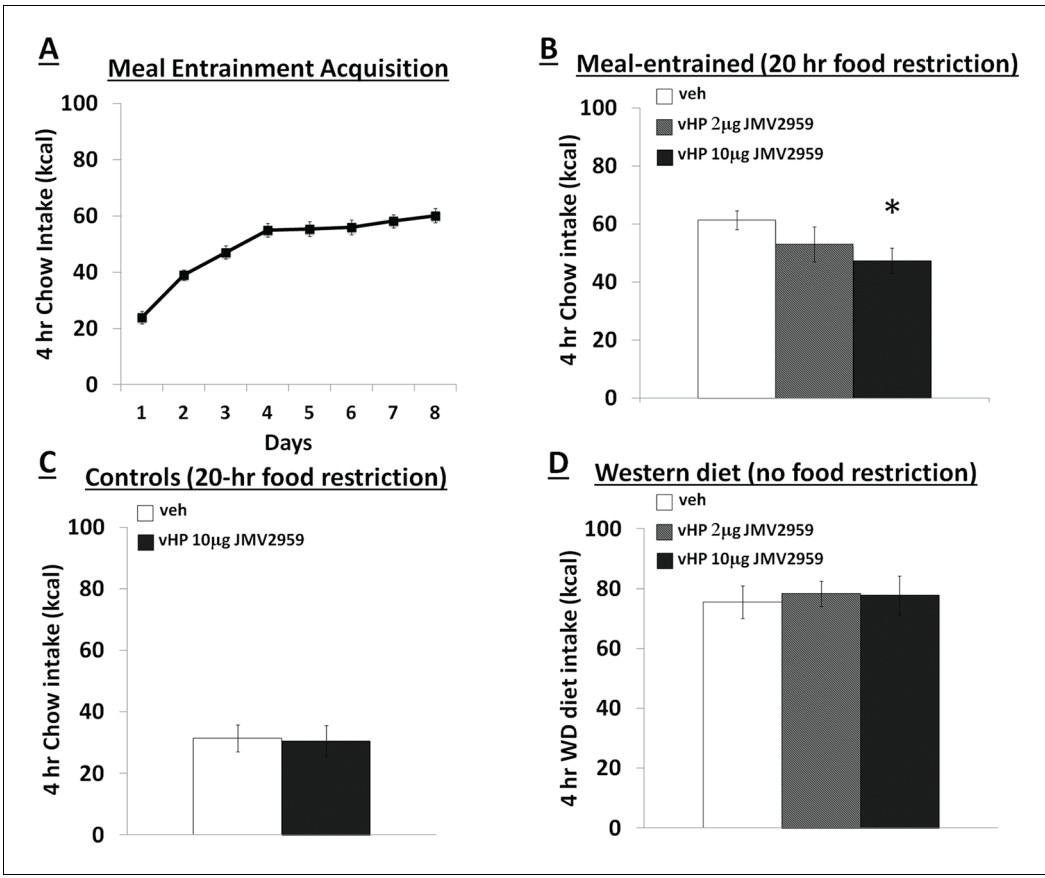

**Figure 2.** Effects of vHP GHSR blockade on conditioned food intake. (**A**) Daily 4 hr chow intake during meal entrainment acquisition, where animals were limited to 4 hr food intake/day for 8 days. (**B**) 4 hr chow intake in meal-entrained animals following bilateral vHP administration of JMV2959. (**C**) 4 hr chow intake in non-meal entrained, 20 hr food-restricted rats following bilateral vHP administration of JMV2959. (**D**) 4 hr Western diet intake in non-meal entrained, non-food deprived rats following bilateral vHP administration of JMV2959. Data are mean ± SEM; *p<0.05.

The following figure supplements are available for Figure 2:

**Figure supplement 1.** Effects of LHA GHSR blockade on conditioned food intake.

meal-entrained rats (*Figure 2D*). Collectively these findings indicate that endogenous vHP ghrelin signaling is critical in the control of conditioned feeding behavior.

## Endogenous ghrelin signaling in LHA neurons does not regulate conditioned feeding behavior

Like the vHP, the LHA is another brain region that expresses GHSR, and activation of these receptors increases feeding (*Mani et al., 2014*; *Olszewski et al., 2003*), Moreover, orexin-producing neurons in the LHA are activated by external food-related cues and by LHA administration of ghrelin (*Olszewski et al., 2003*; *Petrovich et al., 2012*). Thus, to determine whether endogenous LHA ghrelin signaling also contributes to conditioned feeding responses, we repeated the meal entrainment and non-entrained, 20 hr food deprivation procedures in LHA cannulated rats. Animals restricted to 4 hr chow access per day again learned to increase their chow consumption over the 8 day training period (significant main effect of Day, p<0.01) (*Figure 2—figure supplement 1A*). Bilateral delivery of JMV2959 had no effect on 4 hr chow intake in meal entrained rats, nor did LHA JMV2959 application influence 4 hr chow intake in 20 hr food deprived, non-meal entrained rats (*Figure 2—figure*

*supplement 1B,C*). These findings suggest that unlike vHP ghrelin signaling, LHA ghrelin signaling does appear to be critical for the manifestation of conditioned feeding behavior.

## GHSR-expressing vHP CA1 neurons provide input to dpLHA neurons

A vHP to LHA pathway hypothesized to play a role in GHSR signaling for the control of conditioned feeding behavior, was investigated using retrograde pathway-tracing and immunohistochemistry. Following injection of either Fluorogold or cholera toxin B subunit (CTB) into the dpLHA, abundant exclusively ipsilateral retrograde labeling was seen in the vHP field CA1 and subiculum (i.e., no Fluorogold- or CTB-immunoreactive cells were observed in the ventral CA1 contralateral to tracer injections), consistent with previous findings (*Hahn and Swanson, 2010*; *Hahn and Swanson, 2012*; *Cenquizca and Swanson, 2006*). Further investigation of CTB and Fluorogold retrogradely-labeled vHP neurons found that most were also immunoreactive for GHSR [representative dpLHA CTB injection site depicted in *Figure 3A*; retrogradely-labeled CTB immunoreactive soma in the vHP depicted in *Figure 3C*; inset location delineated in *Figure 3B*]. Consistent with previous literature revealing extensive GHSR mRNA expression in the vHP CA fields (*Mani et al., 2014*; *Zigman et al., 2006*), our immunohistochemistry analyses confirmed robust GHSR protein expression in vHP CA1 pyramidal neurons (*Figure 3D*). Quantification of the percentage of back-labeled CTB and Fluorogold immunoreactive soma also expressing GHSR revealed 84.11% (SEM = 0.07) colocalization in rats injected with CTB in the LHA, and 86.22% (SEM = 0.03) colocalization in rats that were injected with Fluorogold in the LHA (representative CTB and GHSR co-labeled soma depicted in *Figure 2E*). These data indicate that the vast majority of vHP neurons that provide an ipsilateral input to the dpLHA also express GHSR.

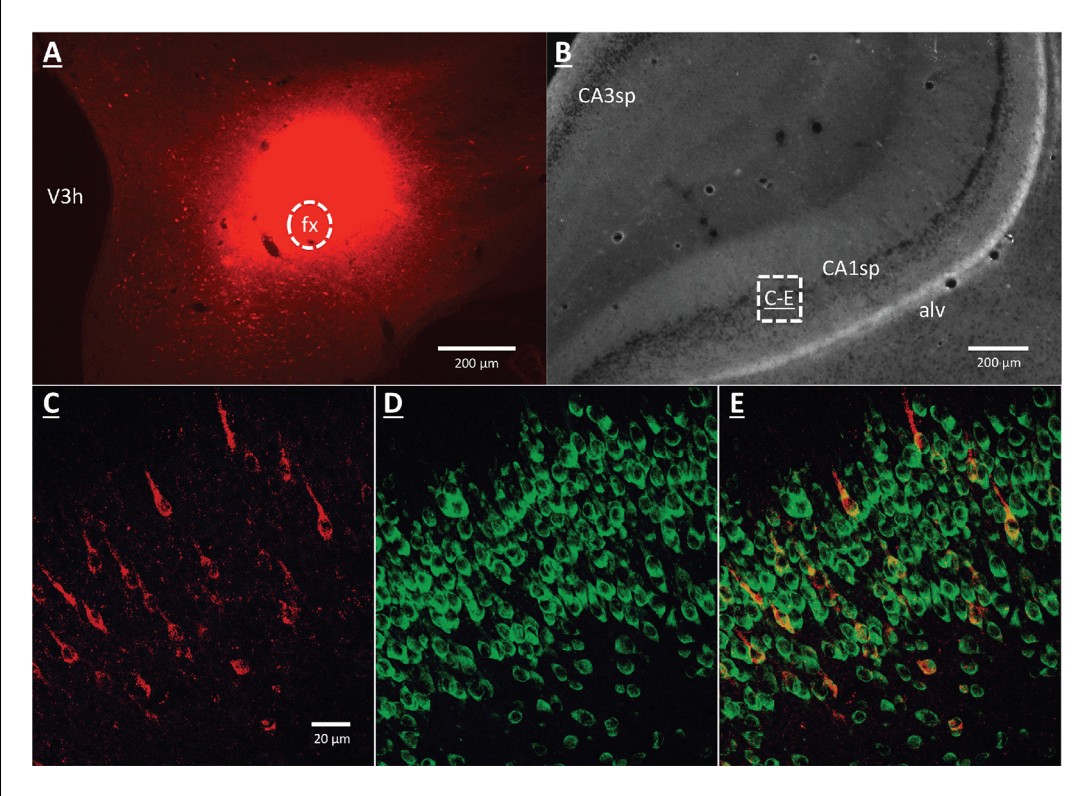

**Figure 3.** GHSR-expressing vHP CA1 neurons provide input to dpLHA neurons. (**A**) A representative dorsal perifornical LHA injection site of CTB-AF594 (red). fx:fornix; V3h: Third ventricle (**B**) Darkfield cytoarchitecture of the vHP, where insets C-E represents field CA1 pyramidal layer (CA1sp) insets from the square (white dashed line). (**C**) vHP retrogradely-labeled CTB immunoreactive perikarya (red). (**D**) GHSR protein expression (green). (**E**) CTB and GHSR co-labeled soma.

## Unilateral LHA lesions block the hyperphagic response following GHSR activation in ipsilateral, but not contralateral vHP neurons

To explore the functional relevance of the exclusively unilateral vHP to dpLHA communication in vHP ghrelin-mediated hyperphagia, rats with unilateral excitotoxic (NMDA) or sham lesions in the dpLHA were implanted with bilateral vHP-targeted cannulae (representative unilateral dpLHA lesion depicted in *Figure 4* with NeuN immunohistochemistry). Food intake was measured after unilateral vHP administration of ghrelin (300 pmol) to rats with dpLHA lesions or sham lesions that were either ipsilateral or contralateral to the side of ghrelin administration (i.e. vHP to LHA communication either eliminated or intact). Ghrelin delivered to the vHP contralateral to the LHA lesion (or sham lesion) significantly increased chow intake in both groups, whereas ghrelin delivered to the vHP ipsilateral to the LHA lesion (or sham lesion) significantly increased food intake only in the Sham group (*Figure 5A–D*; *indicates $p < 0.05$ vs. vehicle treatment, Ŧ indicates $p < 0.07$ vs. vehicle treatment). A repeated-measures overall ANOVA was also performed with 'lesion group' as a between subjects variable and 'drug' as a repeated measures variable. There was no overall significant 'group' effect on food intake at any point, whereas the overall 'drug' effect was significant at all time points ($p < 0.001$). Furthermore, there was a significant interaction between 'group' and 'drug' at the 5 hr time point [$F_{(1,15)} = 4.93$, $p < 0.05$], whereas this interaction just missed significance at the 1 hr and 3 hr time points. These results show that vHP neural communication with the dpLHA is required for vHP ghrelin-mediated feeding effects.

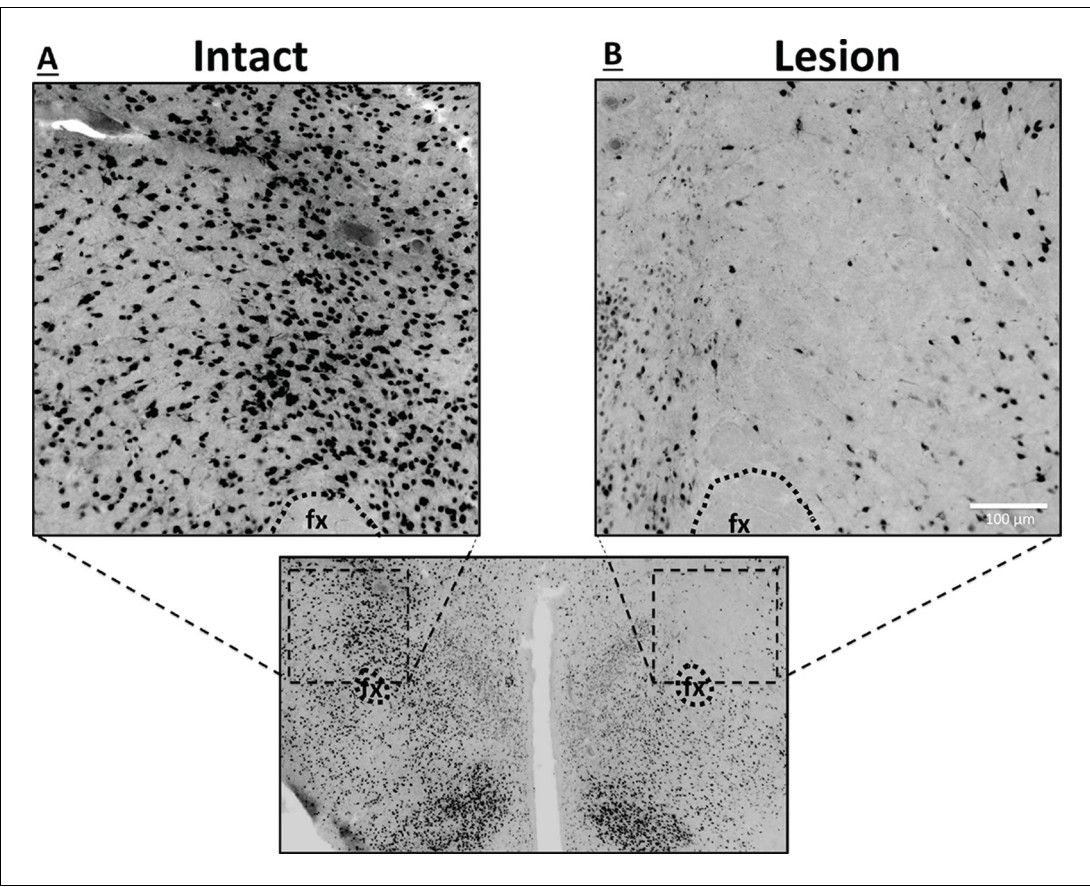

**Figure 4.** Representative unilateral dpLHA lesions. (**A**) NeuN immunohistochemistry contralateral to a NMDA dorsal perifornical (dp) LHA lesion (intact side). (**B**) NeuN immunohistochemistry for a representative unilateral NMDA dpLHA lesion targeting the dorsal perifornical LHA.

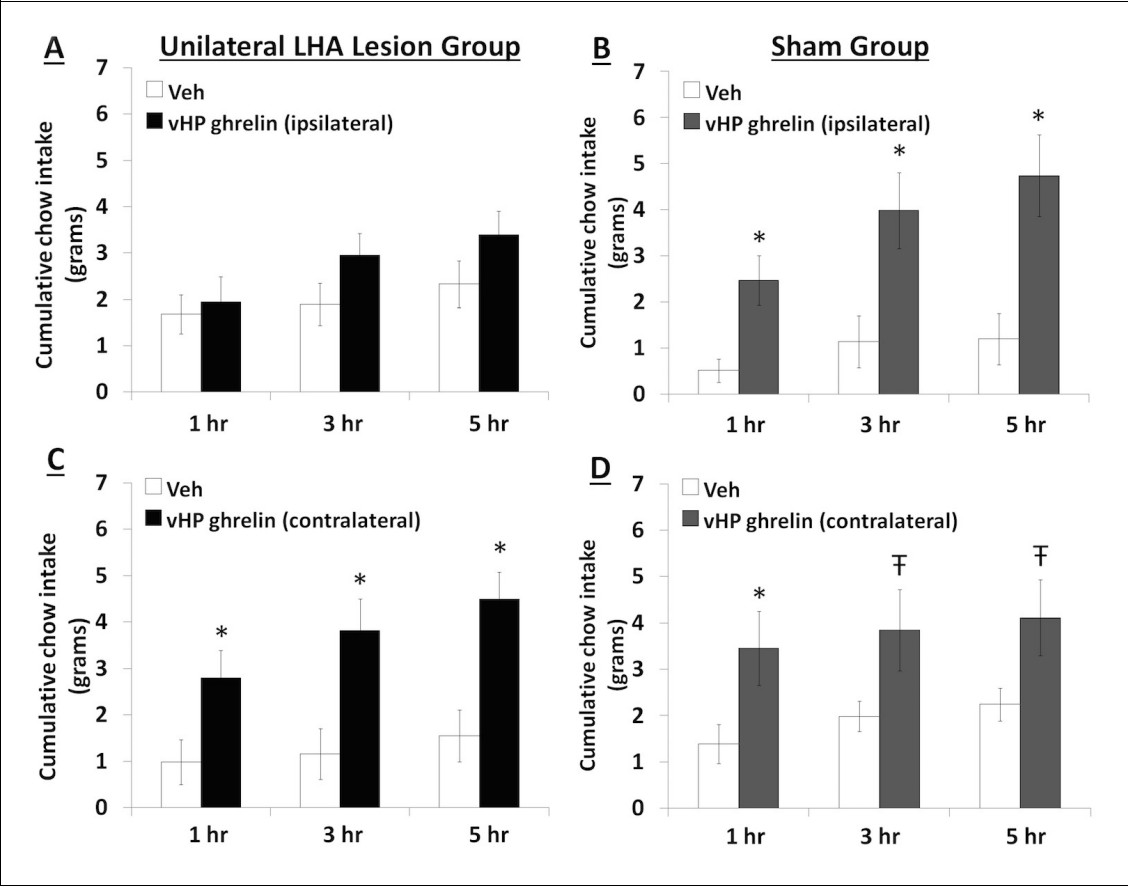

**Figure 5.** Effects of vHP-dpLHA unilateral disconnection on vHP ghrelin-mediated hyperphagia. Cumulative chow intake following unilateral vHP injections of ghrelin (300 pmol) that was: (A) Ipsilateral to a unilateral dpLHA lesion, (B) ipsilateral to a dpLHA sham lesion. (C) contralateral to a dpLHA lesion, and (D) contralateral to a dpLHA sham lesion. Data are mean ± SEM; *p<0.05; Ŧ p<0.07.

## vHP CA1 neurons provide input to dpLHA ORX-expressing neurons

Anterograde pathway-tracing with *Phaseolus vulgaris* leucoagglutinin (PHAL) provided further support for a direct and exclusively ipsilateral connection from the vHP to the dpLHA (*Figure 6—figure supplement 1* displays PHAL immunoreactive axons that are exclusively ipsilateral to a vHP PHAL iontophoresis injection). PHAL was injected by iontophoresis into the vHP field CA1 (representative injection site depicted in *Figure 6A*). Double-label immunohistochemistry for PHAL and ORX revealed several instances of PHAL-labeled axons with numerous varicosities in close apposition to ORX-expressing neurons within the dpLHA (representative examples shown in *Figure 6C*, inset location delineated in *Figure 6B*). Subsequent triple-label immunohistochemistry for PHAL, ORX, and the presynaptic marker synaptophysin indicated that the apparent terminal appositions were putative sites of synaptic input from vHP CA1 neurons to ORX-expressing neurons (representative examples shown in *Figure 6D*). These data confirm an input to the dpLHA from the vHP field CA1, and indicate further that this input targets ORX-expressing neurons.

## vHP GHSR activation increases expression of Fos in dpLHA ORX-expressing neurons

Unilateral vHP ghrelin (300 pmol) administration robustly increased expression of the neural activation marker Fos in the ipsilateral LHA compared to vehicle treatment (p<0.01 vs. vehicle and vs. contralateral LHA). Notably, vHP ghrelin injection activated approximately 70% of orexin-expressing neurons in the ipsilateral LHA (*Figure 7*; p<0.01 compared to vehicle treatment). These data are

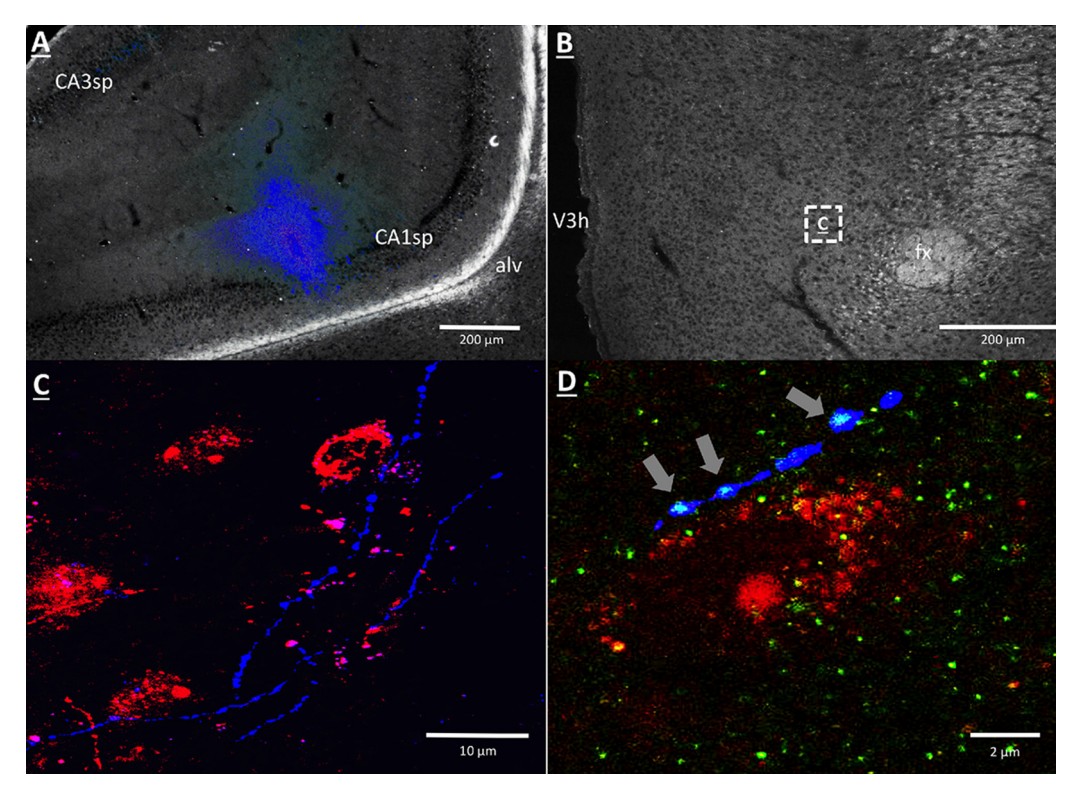

**Figure 6.** vHP CA1 neurons provide input to dpLHA ORX-expressing neurons. (**A**) A representative PHAL (blue) injection site centered in ventral hippocampus field CA1. CA1sp, CA3sp: CA1 and CA3 pyramidal cells. (**B**) Darkfield microscopy, where neuroanatomical analyses were performed in the dpLHA (fx: fornix; V3h: Third ventricle; location shown in C). Inset (**C**) confocal imaging reveals some orexin neurons (red) in the dpLHA that are in close apposition to vHP field CA1-originating axons, labeled with PHAL (blue). (**D**) Triple-label imaging reveals co-labeling of PHAL and synaptophysin (green) in axons in apposition to dpLHA orexin-expressing soma (indicated by white arrows).

The following figure supplements are available for Figure 6:

**Figure supplement 1.** CA1 vHP neurons project ipsilaterally to the dpLHA.

---

consistent with the hypothesis that vHP ghrelin signaling activates orexin-expressing dpLHA neurons.

## Central blockade of Ox1R eliminates vHP ghrelin-mediated hyperphagia

To test whether vHP ghrelin-mediated hyperphagia requires downstream Ox1R signaling, rats first received a lateral ventricle (LV) injection of either vehicle (DMSO) or the Ox1R antagonist, SB334867, at a dose (30 nmol) that was subthreshold for intake effects alone under these conditions. Immediately afterwards, the rats received unilateral parenchymal vHP injections of ghrelin (300 pmol) or vehicle on separate test days. Treatments were counter-balanced using a within-subjects design. Ghrelin significantly increased chow intake in rats pretreated with vehicle in the LV, but had no effect relative to vehicle on intake in rats pretreated with SB334867 (*Figure 8A,B*; *indicates $p < 0.05$ vs. vehicle treatment, Ŧ indicates $p < 0.07$ vs. vehicle treatment). These data indicate that vHP-mediated feeding effects involve downstream Ox1R signaling.

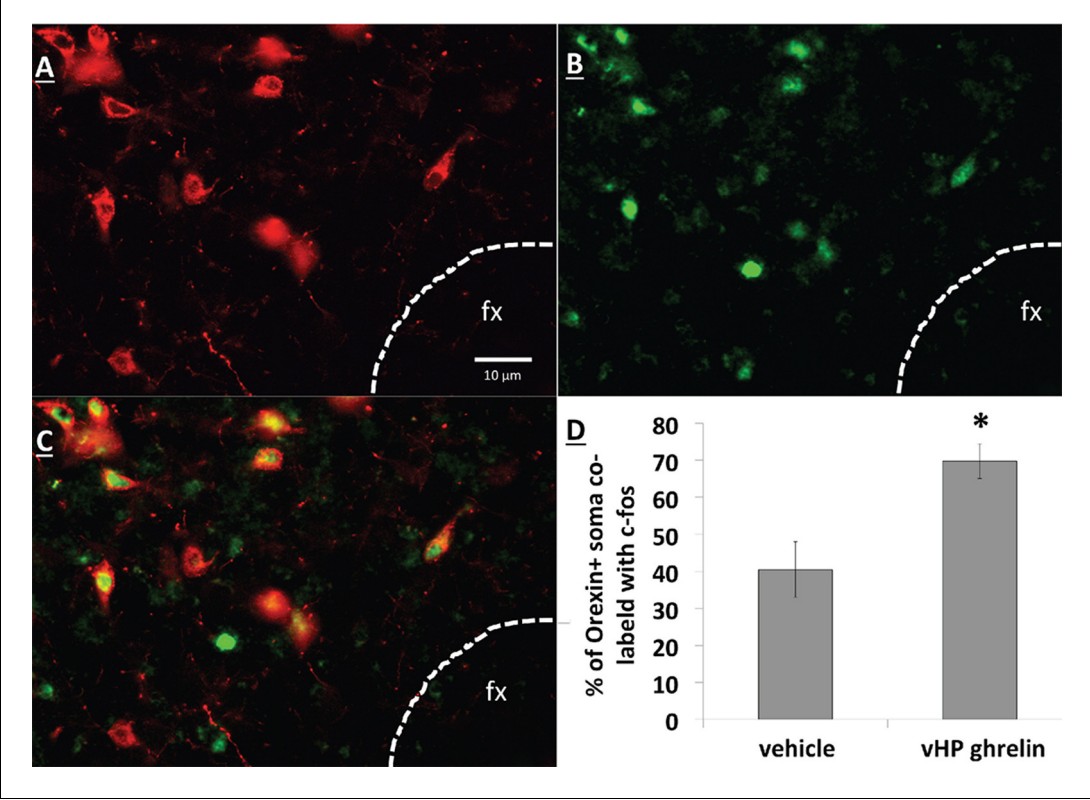

**Figure 7.** vHP GHSR activation increases Fos expression in dpLHA orexin-expressing neurons. (**A**) Orexin-expressing neurons (red) in the dpLHA; fx: fornix. (**B**) Fos expression (green) in the dpLHA induced by ipsilateral vHP ghrelin (300 pmol) administration. (**C**) Co-labeling of dpLHA orexin expression and Fos expression. (**D**) Quantification of orexin positive soma co-labeled with Fos, where co-labeling was significantly higher compared to vHP vehicle injections. Data are mean ± SEM; *p<0.05.

## Discussion

Ghrelin levels rise in anticipation of feeding (*Frecka and Mattes, 2008*; *Cummings et al., 2001*; *Drazen et al., 2006*) and ghrelin receptor null mice show blunted meal anticipatory responding under meal entrainment feeding schedules (*Blum et al., 2009*; *Davis et al., 2011*). Compared to equivalent levels of food restriction, the anticipation of eating in meal entrained rats had a greater stimulatory effect on ghrelin release (*Drazen et al., 2006*), indicating that ghrelin acts as a meal anticipation hormonal signal that facilitates the consumption of large quantities of food. This notion is also consistent with data showing a role for ghrelin in stimulating gastric motility (*Masuda et al., 2000*; *Peeters, 2003*) and increasing meal size (*Kanoski et al., 2013*; *Lee et al., 2011*; *Swartz et al., 2014*; *de Lartigue et al., 2010*; *Faulconbridge et al., 2003*). Here we identify and characterize a novel neural substrate extending these previous findings by demonstrating that vHP ghrelin signaling is physiologically relevant for conditioned feeding behavior. After exposure to a meal entrainment schedule in which rats were restricted to 4 hr food access each day, pharmacological blockade of vHP GHSR signaling prior to food access significantly reduced 4 hr chow consumption. Comparatively, vHP GHSR blockade had no effect on a 4 hr chow intake in equally food-restricted rats that had not undergone the meal entrainment learning. Moreover, the intake reduction appears to be specific to conditioned feeding behavior as opposed to being based on higher baseline kcal consumption in meal entrained vs. control rats. This is supported by results showing that vHP JMV2959 administration did not reduce Western diet intake in non-food deprived, non-meal entrained animals under conditions where baseline kcal intake of the Western diet was actually higher than in meal-entrained rats consuming chow. However, another study showed that peripheral GHSR antagonist (Compound 26) delivery failed to suppress feeding in meal-entrained mice

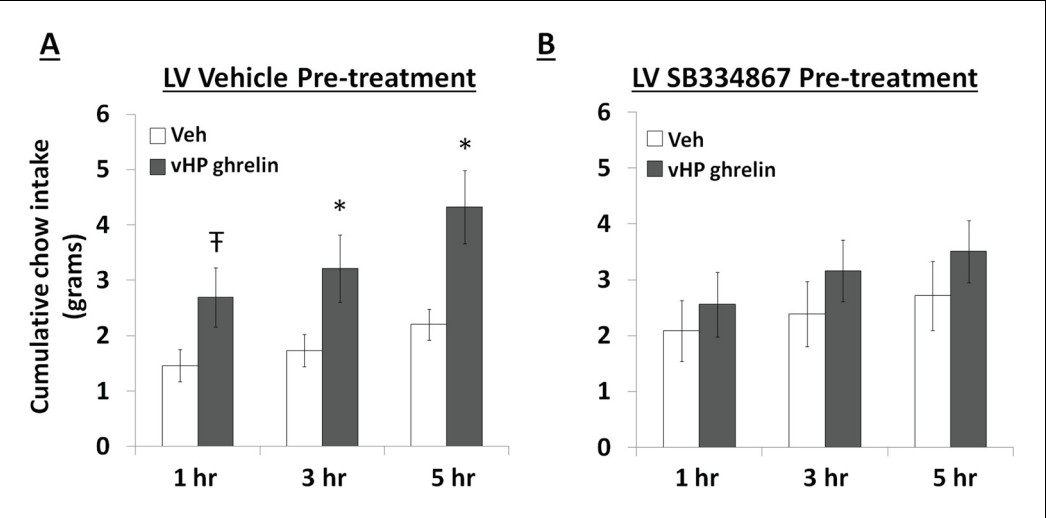

**Figure 8.** Effects of Orexin-1 receptor blockade on ghrelin-mediated hyperphagia. Cumulative chow intake following: (A) pre-treatment with lateral ventricle (LV) vehicle followed by vHP ghrelin (300 pmol, unilateral) administration or (B) LV Ox1R antagonist, SB334867 (30 nmol), followed by vHP ghrelin administration. Data are mean ± SEM; *p<0.05; Ŧ p<0.07.

(*Perello et al., 2010*), yet we note that this previous study utilized a different GHSR antagonist, a different species, and a different route of administration (peripheral). Collectively, these findings build on previous work showing that pharmacological vHP GHSR activation increases meal frequency and size (*Kanoski et al., 2013*). Here we establish the endogenous relevance of vHP GHSR signaling to a specific component of feeding – conditioned feeding. We hypothesize that pharmacologically induced hyperphagia following vHP ghrelin injections occurs, at least in part, by augmenting the saliency of conditioned food-relevant interoceptive stimuli and/or external cues that are present in the environment.

To investigate neural pathways downstream to vHP ghrelin-mediated feeding behavior, we focused on the dpLHA region that contains abundant ORX-expressing neurons (*Swanson et al., 2005*; *Hahn, 2010*), and provide evidence of an input to dpLHA ORX-expressing from GHSR-expressing neurons in the vHP (field CA1). Our data confirm the existence of an exclusively ipsilateral direct input to the dpLHA from the vHP field CA1, and further demonstrate that ~85% of vHP field CA1 neurons retrogradely labeled from the dpLHA are immunoreactive for GHSR. These findings suggest that vCA1 ghrelin-responsive neurons send a direct input to the dpLHA, thereby highlighting the dpLHA as a downstream target for vHP ghrelin-mediated feeding.

Based on research showing an extensive population of orexin producing neurons in the dpLHA (*Swanson et al., 2005*; *Hahn, 2010*; *Sakurai et al., 1998*; *Chen et al., 1999*), as well as functional and neuroanatomical interactions between ghrelin and Ox1R signaling (*Cone et al., 2014*; *Olszewski et al., 2003*; *Perello et al., 2010*; *Solomon et al., 2007*; *Toshinai et al., 2003*), we next investigated the relationship between vCA1 and dpLHA orexin neurons. A direct connection from the vHP to dpLHA neurons was confirmed by the presence of PHAL-immunoreactive axons in the dpLHA following vHP field CA1 PHAL injections. Several instances of PHAL-labeled axons with varicosities and terminal boutons were recorded in close apposition to dpLHA ORX-expressing neurons. Triple-labeling for PHAL, ORX and the presynaptic marker synaptophysin indicated that some of these close appositions were putative sites of synaptic input. Next, we tested whether vHP GHSR activation engages dpLHA orexin neurons. Immunohistochemical analysis of Fos and orexin expression in the dpLHA in rats treated with vHP ghrelin or vehicle injections revealed Fos expression in approximately 70% of dpLHA orexin-expressing neurons. Taken together, these findings indicate a functional pathway involving GHSR-expressing vHP field CA1 neurons and their downstream communication to dpLHA orexin neurons.

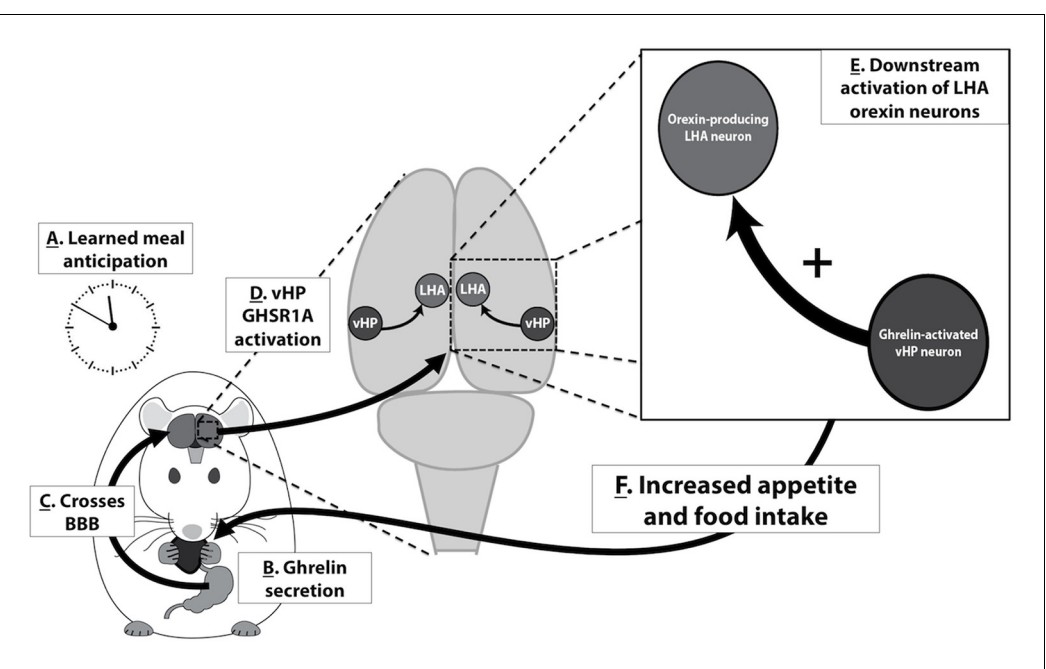

**Figure 9.** A model for vHP ghrelin-mediated conditioned feeding behavior. (A) Animals learn to anticipate a meal based on external or internal food cues. (B,C) Ghrelin secreted from the stomach in anticipation of feeding crosses the blood brain barrier and enters the central nervous system. (D) Ghrelin acts on GHSR in vHP neurons, which engages (E) Downstream activation of dpLHA orexin neurons and CNS Ox1R signaling. (F) Activation of this neural pathway increases conditioned appetite and feeding behavior.

To investigate the functional relevance of a putative vHP field CA1 ghrelin to LHA orexin connection further, we combined behavioral neuropharmacology (vHP ghrelin injection) with an excitotoxic unilateral LHA lesion-mediated neural disconnection approach (adapted after [*Petrovich et al., 2002*]). This approach takes advantage of an exclusively ipsilateral connection between the vHP and the LHA. Results show that vHP ghrelin administration ipsilateral to unilateral LHA lesions (vHP GHSR to LHA communication eliminated) blocks vHP ghrelin-activated hyperphagia that occurs in rats with sham lesions, and that also occurs when vHP ghrelin injections are given contralateral to the unilateral dpLHA lesion (vHP GHSR to LHA communication intact). These data indicate that vHP to dpLHA communication is critical for vHP ghrelin-mediated feeding effects.

A limitation in interpreting these findings with regard to the control of conditioned feeding behavior is that these experiments utilized pharmacological activation of vHP ghrelin receptors during free feeding conditions, and therefore the endogenous relevance of this neural circuitry during conditioned or entrained feeding requires further research. However, our data from Exp. 1 provide strong support for the endogenous relevance of vHP ghrelin signaling system for conditioned food intake. We hypothesize that the robust hyperphagic response following vHP GHSR activation during free feeding conditions is based, at least in part, on pharmacologically activating conditioned feeding behavior in the presence of food-associated cues. In other words, pharmacological vHP GHSR activation is potentially bolstering the saliency of internal (e.g., circadian, metabolic) or external cues within the animal's environment (e.g., food hopper) that have the capacity to promote feeding behavior based on learned associations with feeding. Consistent with this framework, vHP GHSR activation increases meal frequency in response to external environmental stimuli (auditory, visual) that were associated with palatable food access, whereas vHP ghrelin signaling does not augment feeding in response to control stimuli with no prior food association (*Kanoski et al., 2013*). However, because present data do not directly assess the functional importance of vHP ghrelin to LHA orexin neural circuitry in conditioned feeding, we acknowledge that this pathway might also mediate aspects of non-conditioned appetite and feeding behavior. Indeed, ghrelin signaling in other brains regions appears to stimulate non-conditioned appetitive responding (*Skibicka et al., 2011*;

*Skibicka et al., 2013*). For example, ICV or intra-amydala administration of the same GHSR antagonist used in this study (JMV2959) reduces free feeding in fasted (non-entrained) animals (*Salomé et al., 2009*; *Alvarez-Crespo et al., 2012*).

Neurons in the LHA also express GHSR, and LHA orexin-producing neurons are activated by external food-related cues as well as by direct LHA ghrelin administration (*Olszewski et al., 2003*; *Petrovich et al., 2012*). These findings suggest that in addition to the vHP, the LHA might also be a target for ghrelinergic conditioned feeding responses. However, our present data reveal that intra-LHA JMV2959 (including a dose that was effective in the vHP) had no effect on food intake in meal-entrained and non-meal-entrained animals, suggesting that while ghrelin signaling in various brain regions stimulates non-conditioned feeding behavior, vHP ghrelin signaling is a potentially unique neural substrate for learned aspects of feeding behavior. We note that it is also possible that vHP ghrelin to LHA orexin neural circuitry mediates non-conditioned aspects of feeding, however, this notion requires further investigation.

Our neuroanatomical data are consistent with previous research showing that vHP output to the LHA primarily targets the dpLHA region (*Hahn and Swanson, 2010*; *Hahn and Swanson, 2012*; *Kishi et al., 2000*) that was targeted with our unilateral NMDA lesions. While the anatomical and behavioral data in the current study suggest that vHP neurons communicate ghrelinergic signaling to the dpLHA directly, we recognize that multisynaptic pathways are also likely be involved. In addition to providing an input to the LHA, vHP field CA1 neurons also provide an input to several other feeding-related brain regions, including the amygdala and medial prefrontal cortex (mPFC) (*Ishikawa and Nakamura, 2006*; *Cenquizca and Swanson, 2007*). These regions also provide a topographically organized input to the LHA (*Hahn and Swanson, 2015*; *Hahn and Swanson, 2012*), and neural connectivity between the amygdala-LHA and mPFC-LHA has been shown to be critical for conditioned cue-potentiated feeding (*Petrovich et al., 2002*; *Petrovich et al., 2005*). Additional studies are required to determine the functional relevance of these other pathways to vHP ghrelin-mediated appetite.

We examined the requirement of downstream Ox1R signaling for vHP ghrelin-mediated feeding effects, demonstrating that central Ox1R blockade (via LV injections of an Ox1R antagonist) attenuated the hyperphagic effects of vHP ghrelin signaling. However, because ventricular injections spread throughout the CNS and Ox1R s are expressed throughout the brain, (*Hervieu et al., 2001*; *Marcus et al., 2001*), we can only speculate as to the mediating sites of Ox1R activation. Recent research has shown that blockade of Ox1Rs in the VTA attenuates food intake induced by LV ghrelin (*Cone et al., 2014*). Given these findings, and the role of VTA dopaminergic signaling in various aspects of feeding behavior (*Cone et al., 2014*; *Skibicka et al., 2013*; *Nieh et al., 2015*; *Liu and Borgland, 2015*; *Narayanan et al., 2010*; *Cacciapaglia et al., 2011*), we hypothesize that VTA Ox1R signaling is a critical downstream target of vHP ghrelin to dpLHA orexin signaling, however, this remains to be tested. Previous literature has also shown that central orexin signaling has roles in sleep/wake and arousal in addition to regulating feeding behavior (*Hagan et al., 1999*; *Chemelli et al., 1999*). The current study did not directly assess the relationship between vHP ghrelin signaling and orexin mediated arousal. It is possible that the enhanced saliency of external food-related cues following vHP ghrelin signaling is mediated by increased arousal, as well as potential changes to circadian activity, from downstream activation of central Ox1R's (possibly in the locus coeruleus). Further studies are required to test these possibilities, as well as the role of downstream Ox1R signaling in other aspects of feeding behavior (*Zhu et al., 2002*; *Goforth et al., 2014*; *Zheng et al., 2007*; *Parise et al., 2011*; *Harris et al., 2005*; *Cason and Aston-Jones, 2013*; *Thorpe and Kotz, 2005*; *Thorpe et al., 2006*).

Based on the present results and the findings of related previous research, we propose the following model in which vHP ghrelin signaling engages conditioned feeding behavior (*Figure 9*): [1] Ghrelin is secreted from the stomach in response to cues (internal or external) related to meal anticipation, [2] Ghrelin crosses the blood brain barrier and enters the central nervous system, [3] vHP GHSR-expressing neurons are activated by ghrelin, [4] These vHP ghrelin-activated neurons engage downstream dpLHA orexin-expressing neurons, [5] Activation of this neural pathway increases appetite and feeding behavior. Together, our data highlight novel neurobiological mechanisms by which ghrelin engages higher-order brain circuitry, that in turn controls hypothalamic systems necessary for the regulation of appetite.

## Materials and methods

### Animals

Male Sprague-Dawley rats (Harlan, Indianapolis, IN) (320–450 g) were individually caged in a temperature-controlled vivarium with unlimited access (except where noted) to water and food (LabDiet 5001, LabDiet, St. Louis, MO ) on either 12 hr:12 hr light/dark cycle (lights on at 8:00 AM) or 12:12 hr reverse light/dark cycle (lights off at 10:00AM). For all surgical procedures, rats were anesthetized (ketamine 90mg/kg and xylazine, 2.8 mg/kg), and sedated (acepromazine. 0.72 mg/kg). All procedures were approved by the Institute of Animal Care and Use Committee, at the University of Southern California.

### Central drug injections, and excitotoxic (NMDA) lesions (Experiments 1, 3, 5 and 6)

Experiments involving central drug injections were as follows: (Exp. 1) bilateral vHP injection of a ghrelin receptor (GHSR) antagonist (JMV2959, EMD Millipore, Billerica, MA); (Exp. 3) unilateral dpLHA injections of NMDA (Sigma-Aldrich, St. Louis, MO); (Exp. 3 and 5) unilateral vHP injection of ghrelin (Bachem, Torrance, CA); (Exp. 6) bilateral vHP injection of ghrelin and lateral ventricle injection of an orexin receptor type 1 (Ox1R) antagonist (SB334867, Tocris Bioscience, Ellisville, MO). Drugs were dissolved in artificial cerebrospinal fluid (aCSF), except SB334867 which was dissolved in DMSO.  Drugs were delivered using a microinfusion pump (Harvard Apparatus, Holliston, MA) connected to a 33-gauge microsyringe injector, via indwelling guide cannulae. Flow rate was calibrated and set to 5 µl/min; injection volume was 100 nl/hemisphere for Exp. 1, 3 and 6. Injection volume for NMDA lesions were 200 nl (dose: 4 µg). Ventricular injections in Exp. 6 were 1 ul in volume. Injectors were left in place for 30 sec post-injection.

Guide cannulae (26-gauge, Plastics One, Roanoke, VA) for bilateral vHP drug injections were implanted at the following stereotaxic coordinates (*Paxinos and Watson, 2007*): -4.9 mm anterior/posterior (AP), 4.8 mm medial/lateral (ML), 6.1 mm dorsal/ventral (DV); they were then affixed to the skull with jewelers screws. Coordinates for lateral ventricular injections of Ox1R antagonist SB334867 were -0.9 mm AP, 1.8 mm ML, 2.6 mm DV. Coordinates for excitotoxic (4 ug, NMDA, 200 nl) LHA lesions, were -2.9 mm AP, 1.1 mm ML, -8.9 mm DV. Injectors for drug administration projected 2 mm beyond guide cannulae. Coordinates for LHA injections were -2.9 mm AP, +/- 1.0 mm ML, -6.7 mm DV. Placement of vHP and LHA cannulae placement was determined post-mortem by injection (via guide cannulae) of a blue dye (100 nl 2% pontamine sky blue ink). Data from animals with dye restricted to the vHP were included in the analysis (a representative vHP injection site is shown in *Figure 1*). In total, 12% of rats with vHP cannulae were excluded from the analysis because of misplaced cannulae, determined by the presence of blue dye as noted above.

The extent of unilateral NMDA dpLHA-targeted lesions was determined postmortem by immunohistochemical detection of the neuronal marker NeuN. Rats showing pronounced (65%) loss of NeuN labeling within the dpLHA compared to a contralateral sham (aCSF) injection in the same animal were included in data analysis (two rats with minimal or misplaced lesions were excluded).

Targeting of the OX1R antagonist SB334867 to the lateral ventricle (Exp. 6) was verified 1 week post-injection of the antagonist by elevation of cytoglucopenia resulting from injection (at the same coordinates) of 210 µg (2 µl) of 5-thio-D-glucose (*Ritter et al., 1981*). A post-injection elevation of at least 100% of baseline glycemia was required for subject inclusion (1 rat was excluded).

### Neural pathway-tracing (Exp. 2 and 4)

Experiments involving neural pathway-tracing were as follows: (Exp. 2) Retrograde tracing from the dpLHA following unilateral pressure injection of either Fluorogold (FG, Fluorochrome LLC, Denver, CO; 2% in 0.9% NaCl), or cholera toxin B subunit conjugated to (red fluorophore) Alexa Fluor 594 (CTB-AF594, 0.25% in 0.9% NaCl, Life Technologies, Carlsbad, CA; Cat. No. C-22842). Pressure injections were performed using a microinfusion pump as described above; injection volumes were 100 nl (Exp. 4). Anterograde tracing from the vHP field CA1 following unilateral iontophoretic injection of *Phaseolus vulgaris*-leucoagglutinin (PHAL, Vector Labs, Burlingame, CA; 2.5% in 0.1 M sodium phosphate-buffered saline, pH 7.4) was performed using a precision current source (digital midgard precision current source, Stoelting) as described previously (*Hahn and*

*Swanson, 2010*). PHAL immunoreactive axons in the LHA confirmed the exclusively ipsilateral nature of vHP to LHA projections (*Figure 6—figure supplement 1*; representative vHP PHAL injection site, *Figure 6*).

## Immunohistochemistry

Experiments 2–5 included immunohistochemical (IHC) analyses. Animals were anesthetized, then perfused transcardially with ice-cooled 0.9% saline, followed by 4% paraformaldehyde in 0.1 M borate buffer of pH 9.5. The brains were removed and immersed in the same (unused) fixative containing 12% sucrose for 20–24 hr at 4°C. The brains were then blocked transversely at the level of the caudal midbrain, and each block was flash-frozen in dry-ice cooled hexane before being sectioned frozen on a sliding microtome (transverse plane, 30 μm thickness, 5 series). Sections were stored in antifreeze solution at -20°C until further processing (Antifreeze solution: 30% ethylene glycol, 20% glycerol in 0.02 M potassium phosphate-buffered saline—KPBS).

The following primary antibodies were used: (Exp. 2) anti-CTB (raised in goat) and anti-GHSR (goat); (Exp. 4) anti-PHAL (goat), anti-ORX1a (rabbit), and anti-synaptophysin (mouse); (Exp. 5) anti-cFos (sheep) and anti-ORX1a (rabbit). Antibody incubations were done at 4°C (washes and other steps at room temperature). Primary antibody incubation was overnight (~18 hrs). Primary antibodies were diluted in the following solution: KPBS containing 1% donkey serum and 0.1% Triton X-100; this step was followed by KPBS washes (8 changes, 10 mins each). Overnight secondary antibody incubation: Biotinylated secondary antibodies or secondary-fluorophore conjugates were diluted in KPBS containing 0.1% Triton X-100; this step was followed by KPBS washes (6 changes, 10 mins each). Overnight tertiary incubation (if biotinylated secondary antibodies were applied: Streptavdin-fluorophore conjugate was diluted in KPBS; this step was followed by KPBS washes (3 changes, 10 mins each). Sections were mounted onto glass slides, allowed to dry, and coverslipped, using 50% glycerol in KPBS mountant. Coverslip edges were sealed with clear nail polish.

Controls for IHC: For each primary antibody used, a standard control was performed in which the primary antibody was omitted from the protocol (to control for non-specific signal resulting from subsequent antibody and signal-detection reagents). In addition, for GHSR primary antibodies, an adsorption / neutralization was performed using the GHSR peptide sequence (a propriety sequence derived from a portion of the complete GHSR sequence). Following this adsorption, GHSR labeling was abolished. Photomicrographs were acquired using either a Nikon 80i and (Nikon DS-QI1,1280X1024 resolution, 1.45 megapixel) under epifluorescent illumination, or as optical slices using a Zeiss LSM 700 UGRB Confocal System (controlled by Zeiss Zen software). All photomicrographs in the figures are oriented such that: up = dorsal, down = ventral, left = medial, right = lateral.

## Experiment 1: Effect of vHP-mediated ghrelin signaling on conditioned feeding behavior

Rats were implanted bilaterally with vHP-targeted cannulae (n=11). After recovery from surgery, they were put on a meal-entrainment schedule where they were limited to food access for 4 hr daily, beginning at 10:00 AM (start of dark cycle), for 8 days. One day after the last training day, immediately prior to lights off, the meal-entrained rats were injected via their bilaterally implanted cannulae with either 0 μg, 1 μg, or 5 μg of a ghrelin receptor (GHSR) antagonist (JMV2959; total doses: 0 μg, 2 μg, 10 μg). Dose selection for JMV2959 was based on previous reports (*Skibicka et al., 2011*; *Alvarez-Crespo et al., 2012*). Treatments were separated by 2–3 intervening day using a counterbalanced within-subjects design. Food intake was recorded 4 hr post-injection (food spillage accounted for). To control for the effects of vHP GHSR antagonism on food intake independent of meal-entrainment, 2 additional groups of rats (n=8/group) were 20 hr food deprived before receiving vHP injections of vehicle or JMV2959 (10 μg) and having their food intake recorded (as described above). A between-subjects design was used to avoid conditioning effects with repeated food restriction. Given the baseline differences in food intake between meal-entrained and control rats, to control for a lack of observable effects based on low kcal intake (i.e., floor effect) in controls, a separate cohort of non-food restricted rats (n=11) were first habituated to 1 ~3 g pellet of a 'Western diet' a day (45% kcal fat, enriched with sucrose; Research Diets D12451) for 5 days, given at random times to avoid conditioning effects. Two days later, animals received vHP injections of 0 μg, 1 μg, or

5 µg JMV2959 (total doses: 0 µg, 2 µg, 10 µg) using a counter-balanced within subjects design (treatments separated by 2–3 intervening days) and food intake was recorded as described above.

To determine whether endogenous LHA ghrelin signaling also contributes to conditioned feeding responses, we repeated the meal entrainment (n=9) and non-entrained, 20 hr food deprivation (n=15) procedures described above with bilateral, intra-LHA injections of JMV2959 (total doses: 0 µg, 2 µg, 10 µg).

### Experiment 2: Investigation of dorsal perifornical LHA (dpLHA) input from vHP GHSR-expressing neurons

Each rat (n=6) received a unilateral injection of CTB and a unilateral contralateral injection of FG targeted to the dorsal perifornical LHA (dpLHA). Seven days later, rats were fixation-perfused, and brains removed and processed as described. Fluorogold was examined using its native fluorescence. Detection of CTB by IHC followed the steps described above, except the following antibodies and fluorophores were used: anti-CTB primary antibody (mouse monoclonal, 1:500, Abcam, Cambridge, MA Catalog #ab62429); biotinylated anti-mouse secondary antibody (1:500, donkey polyclonal, Jackson Immunoresearch, West Grove, PA); streptavidin DyLight 549 tertiary reagent (1:500, Jackson Immunoresearch; overnight). For IHC detection of GHSR the following antibodies were used: Anti-GHSR1 primary antibody (goat, polyclonal; 1:500, Santa Cruz, Santa Cruz, CA, Catalog #sc-10362); anti-goat Alexa Fluor 488 secondary antibody (donkey polyclonal, 1:500, Jackson Immunoresearch).

### Experiment 3: Effects of excitotoxic (NMDA) dorsal perifornical LHA lesions on vHP ghrelin-mediated feeding

Rats received unilateral NMDA injections targeted to the dpLHA (n=10) or sham lesions (n=7) and cannulae targeting the bilateral vHP. 2 hr into the rats light cycle (lights on at 8 AM), 0 pmol or 300 pmol ghrelin was administered to the vHP ipsilateral to the dpLHA lesion. Unilateral ghrelin doses were determined based on pilot studies (data not shown) and previous literature (*Kanoski et al., 2013*), where the maximally effective dose for hyperphagia was chosen. In a separate experiment, the same cohort received 0 pmol or 300 pmol ghrelin to the vHP contralateral to the dpLHA lesion. Treatments were separated by intervening 1 day using a counterbalanced mixed design, with sham or lesion as a between-subjects variable, and drug as within-subjects. Chow intake was recorded at 1, 3, and 5 hr after injections (food spillage accounted for).

### Experiment 4: vHP output to orexin-expressing dpLHA neurons

Rats (n = 4) received iontophoretic injections of PHAL targeting the vHP. Seven days later, the rats were euthanized and tissue was harvested and processed according to procedures outlined in *Materials and methods*. IHC detection of PHAL proceeded according to the above described procedures using goat anti-PHAL primary antibody (1:5000, Vector Labs, Catalog #AS-2224), followed by a donkey anti-goat secondary antibody conjugated to Cy3 (1:500, Jackson Immunoresearch, 4 hr). Material from the experiment with a PHAL injection site that was most restricted to the vHP field CA1 region of interest was selected for subsequent IHC analysis of vHP connection interactions with dpLHA orexin neurons. IHC detection of orexin was performed as described using a rabbit anti-ORX1 primary antibody (1:1000, Phoenix Pharmaceuticals, Burlingame, CA; Catalog #H-003-30, 18 hr), was followed by a donkey anti-rabbit secondary antibody conjugated to AMCA (1:500, Jackson Immunoresearch,).

IHC detection of synaptophysin was performed as described using a mouse anti-synaptophysin primary antibody (1:1000, PROGEN Biotechnik, Heidelberg, Germany; Catalog #61012, 18 hr in room temperature), followed by a donkey anti-mouse secondary antibody conjugated to Alexa Fluor 488 (1:500, Jackson Immunoresearch).

### Experiment 5: Effects of vHP ghrelin injection on Fos expression in orexin-expressing LHA neurons

Rats received either unilateral vHP ghrelin (n=5; 300 pmol) or aCSF vehicle injections (n=4). 60 min later, animals were euthanized and tissue was harvested and processed according to procedures outlined in *Materials and methods*.

Fos was used as a marker for neuronal activity, and IHC detection was performed according to the procedure above using a sheep anti-Fos primary antibody (1:2000, Cell Signaling, Danvers, MA; Catalog # 2250s, 18 hr), followed by a donkey anti-goat secondary antibody conjugated to Alexa Fluor 488 (1:500, Jackson Immunoresearch, 4 hr).

IHC detection of orexin was performed as above using a rabbit anti-Ox1a primary antibody (1:1000, Phoenix Pharmaceuticals, Catalog #H-003-30, 18 hr) and was followed up with an incubation of a donkey anti-rabbit secondary antibody conjugated to Cy3 (1:500, Jackson Immunoresearch).

Quantification of co-localization of Fos and orexin positive cells was calculated by an experimenter blind to the treatment conditions using ImageJ software (cell count Plugin) based on the number of co-localized Fos and orexin cells divided by the total number of orexin positive cells.

## Experiment 6: Effect of introcerebroventricular injection of an OX1R antagonist on vHP ghrelin induced feeding.

Rats (n=15) received cannulae targeting the both the vHP and LV. 2 hr into light cycle, animals received LV injections of 0 nmol or 30 nmol the orexin-1 receptor antagonist, SB334867, immediately followed by 0 pmol or 300 pmol ghrelin to the vHP. Dose of SB334867 was determined based on previous literature (*Parise et al., 2011*; *Haynes et al., 2000*). Treatments were separated by 1 intervening day using a counterbalanced within-subjects design. Chow intake was measured 1, 3, and 5 hr after injections (food spillage accounted for).

### Statistical analysis

All statistical analyses employed repeated measures or one-way analysis of variance (ANOVA). The $\alpha$ level for significance was 0.05. Statistical analyses were conducted with computer software (Statistica V7; Statsoft). To ensure the minimum sample size required to be reasonably likely to detect the hypothesized effect, power analyses were conducted for each proposed experiment based on our pilot and recently published experiments using Statsoft software (Statistica 64, V7). Alpha level was set at 0.05 for power analyses; the Root Mean Square Standardized Effect (RMSSE) was estimated separately for each experiment

## Acknowledgements

We thank the following individuals for notable contributions to this work: Dr. Alan G Watts, Dr. Larry W Swanson, Dr. Harvey J Grill, Ashley Lam, Lilly Taing, Hrant Gevorgian, Joanna Liang, Ryan Usui, Mehul Trivedi, Agustina Kim, Allison Apfel, Kaitlin Sontag, and Natalie Demirjian. This study was supported by the National Institute of Health grants DK097147, DK102478, and DK104897 (SEK) and pilot grant funding from the USC Diabetes and Obesity Research Institute (SEK and JDH). We report no biomedical financial interests or potential conflicts of interest.

## Additional information

### Funding

| Funder | Grant reference number | Author |
|--------|------------------------|--------|
| National Institute of Diabetes and Digestive and Kidney Diseases | DK097147 | Scott E Kanoski |
| University of Southern California | USC Diabetes and Obesity Research Institute Pilot Grant | Joel D Hahn Scott E Kanoski |
| National Institute of Diabetes and Digestive and Kidney Diseases | DK102478 | Scott E Kanoski |
| National Institute of Diabetes and Digestive and Kidney Diseases | DK104897 | Scott E Kanoski |

The funders had no role in study design, data collection and interpretation, or the decision to submit the work for publication.

## Author contributions

TMH, JDH, EEN, ANS, SEK, Conception and design, Acquisition of data, Analysis and interpretation of data, Drafting or revising the article; VRK, Acquisition of data, Analysis and interpretation of data, Drafting or revising the article; JT, EMN, Acquisition of data, Analysis and interpretation of data

## Ethics

Animal experimentation: This study was performed in strict accordance with the recommendations by the Institute of Animal Care and Use Committee at the University of Southern California. All animals were handled according to approved IACUC protocols (#11963, 20283, 12009, 20165) of the University of Southern California. For all surgical procedures, rats were anesthetized (ketamine 90mg/kg and xylazine, 2.8mg/kg), and sedated (acepromazine. 0.72mg/kg) and every effort was made to minimize suffering.

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
