## [Decision Letter]

Thank you for submitting your work entitled "Hippocampus ghrelin signaling mediates conditioned appetite through lateral hypothalamic orexin pathways" for peer review at *eLife*. Your submission has been favorably evaluated by a Senior editor, a Reviewing editor, and two reviewers.

The reviewers have discussed the reviews with one another and the Reviewing editor has drafted this decision to help you prepare a revised submission.

This paper describes a neural circuit from ventral hippocampus to the orexin neurons in the lateral hypothalamus that controls conditioned feeding responses.

Both reviewers indicated considerable enthusiasm for this paper; however they both suggested a few changes that would enhance its impact. The most critical concern is that the title and beginning of Results indicate that hippocampal ghrelin signaling is important for 'conditioned feeding' but not for ad lib feeding or feeding after a fast. The title also implicates the circuit from the hippocampus to the orexin neurons in the lateral hypothalamus as being involved in 'conditioned feeding' but those studies did not use the conditioned feeding paradigm. The authors need to include data for the 'conditioned feeding' paradigm in their experiments #3 and #6 if they want to make the claims proposed in title and Abstract. If the authors want to broaden the role of ghrelin acting on the 'hippocampus to orexin neuron circuit' to mediate many aspects a feeding, then they should compare the relative importance of hippocampal input to orexin neurons with other potential inputs (reviewer #2 comment 3). The authors should also address the other concerns raised by the reviewers.

Minor comment from the Reviewing editor: It is customary to put a space between numbers and units, e.g. 4 hr, not 4 hr. Also, abbreviations should be used consistently, e.g. in the first paragraph of Results, we see 4 hr and 4h food intake. Finally, there is only one Fos gene in the rat, so there is no need to distinguish c-Fos from v-Fos (which is of viral origin); i.e., Fos is sufficient.

*Reviewer #1:*

In this study entitled "Hippocampus ghrelin signaling mediates conditioned appetite through lateral hypothalamic orexin pathways", by Ted M Hsu and colleagues describe a new neuronal circuit mediating ghrelin-induced eating, with a focus on the effects of ghrelin on meal-entrained food intake. Using several classical neuroanatomical techniques, including histochemistry, targeted stereotaxic injections, anterograde and retrograde tracing, and brain site lesioning in rats, the investigators now describe a pathway in which GHSR-expressing ventral hippocampal neurons from the CA1 region project to ipsilateral dpLHA orexin neurons. When activated by ghrelin, this pathway results in ghrelin-induced food intake.

This paper is strong for several reasons:

1) Theme: While the hippocampus has been described before as playing a role in various aspects of ghrelin action, the neuronal circuits within which ghrelin-responsive hippocampal neurons participate have not been known. This study describes for the first time a ghrelin-responsive neuronal circuit involving the hippocampal neurons.

2) Technique: The neuroanatomical techniques utilized are classic and performed elegantly.

A major comment follows, although it should not require any major changes to address: The investigators frame this work and this pathway as all being relevant to ‘conditioned feeding’ which they model with about a week of restricted access to food for only 4 hr per day. However, I think it is premature with the data provide to attribute this pathway as being relevant specifically to their model of conditioned feeding. Only in the experiments summarized in Figure 2 was the meal entrainment done – and for those experiments, injections of a GHSR antagonist directly into the vHP were used to confirm that GHSR-expressing vHP neurons were important for conditioned feeding (but not for 4 hr rebound feeding following a 20 hr food restriction). In the subsequent experiments, neuroanatomic techniques were used to track the projections of GHSR-expressing/ghrelin-responsive neurons to the LHA. Also, the orexigenic effects of pharmacologic ghrelin-injections into the vHP were assessed in LHA-lesioned animals or Orexin receptor blocker-administered animals. Activation of LHA neurons was also assessed in animals administered ghrelin into the vHP. Thus, while GHSR-expressing vHP neurons appear to be important for conditioned feeding, the GHSR-expressing vHP neurons projecting to the LHA orexin neurons are not necessarily involved in that particular aspect of ghrelin-induced eating. Thus, the possible role of this newly described pathway in a broader scope of ghrelin's many feeding-related actions should be mentioned. If the investigators want to limit this pathway to conditioned feeding, further physiologic experiments would be needed – e.g., inhibit downstream portions of that pathway and determine how it affects conditioned feeding.

*Reviewer #2:*

The set of studies presented here are important in addressing the neurobiological substrates of overeating in today's society – where consumption is rarely triggered by caloric deficit. They are also important in elucidating an under-appreciated node for the regulation of feeding – the vHP. The work is well conducted and thorough. The authors provide ample data to support most of their conclusions. I have three major comments that need to be addressed:

1) A key argument, and the title of the paper, posits that ghrelin action in the ventral hippocampus augments the conditioned aspects of feeding through LH orexin neurons. A limitation of the manuscript is that the authors do not actually show this. First, they show that hippocampal ghrelin augments conditioned bouts of feeding. They then go on to show that hippocampal ghrelin increases unconditioned food intake and this requires an intact ipsilateral LH (Figure 5) and the availability of central orexin receptors (Figure 8). However, the authors do not show that central orexin receptors are required for ventral hippocampal ghrelin to augment conditioned feeding. Repeating the approach in Figure 1 but infusing either vehicle or the SB334867 into the lateral ventricle would solidify the argument. The use of an orexin saporin for a selective LH lesion would also solidify the argument.

2) In Figure 5, the authors compare food intake induced by vHP ghrelin (or vehicle) in animals with (or without) LH lesions. They compare these groups separately according to whether or not the animals had an LH lesion or sham (Figure 5). These data should be combined and compared using an ANOVA with main effects of surgery (Lesion, Sham) and treatment (Vehicle, Ghrelin). This comparison will help parse out any non-specific effects of the LH lesion unrelated to ghrelin treatment. Moreover, this is critical because bilateral LH lesions are well known to alter feeding behavior and it appears there are differences in the amount of food consumed between the lesion and sham animals that received vehicle. For example, compare food intake vehicle treated animals in lesion and sham groups at 1 hr and 5 hr.

3) The authors are very thoughtful in their Discussion. However, if the authors posit that the source for vHP GHSR activation is peripheral ghrelin evoked by interoceptive or external cues then GHSRs are likely to be activated in other brain regions where they are expressed – including LHA GHSRs. LHA orexin neurons express GHSRs and are activated by cues. While the authors are to be commended for showing that vHP GHSRs are required for the full expression of conditioned feeding in the present paradigm – supporting an endogenous roll for ghrelin in the vHP, the manuscript would be strengthened by investigating another GHSR-rich region to determine if similar effects are observed or whether the GHSRs in the vHP uniquely participated in meal-entrained conditioned feeding. At the very least, this unresolved issue should be discussed.

---

## [Author Response]

*[…] Both reviewers indicated considerable enthusiasm for this paper; however they both suggested a few changes that would enhance its impact. The most critical concern is that the title and beginning of Results indicate that hippocampal ghrelin signaling is important for 'conditioned feeding' but not for ad lib feeding or feeding after a fast. The title also implicates the circuit from the hippocampus to the orexin neurons in the lateral hypothalamus as being involved in 'conditioned feeding' but those studies did not use the conditioned feeding paradigm. The authors need to include data for the 'conditioned feeding' paradigm in their experiments #3 and #6 if they want to make the claims proposed in title and Abstract.*

We thank the reviewers for bringing this concern to our attention. We acknowledge the possibility that the hippocampal ghrelin to lateral hypothalamic orexin circuit could potentially be involved in aspects of appetite and feeding behavior other than conditioned feeding. We further agree that the data presented in Exp. 3 and 6 do not directly assess the importance of this circuit in conditioned feeding. To address this issue, we considered repeating the conditioned feeding paradigm using designs parallel to Exps. 3 and 6, however, we note that the animals are robustly hyperphagic during 20 hr food restricted meal entrainment conditions, and thereby activation of vHP ghrelin receptors would be unlikely to augment intake from a high baseline of excessive food intake (i.e., ceiling effect). Moreover, an antagonist approach for the ghrelin receptor in Exps. 3 and 6 is not logical since these studies are designed to investigate downstream pathways.

Nevertheless, we agree that our previous manuscript draft did not sufficiently consider a role for vHP ghrelin in other (e.g., non-conditioned) aspects of appetite and feeding. Accordingly, we have revised the title and Abstract, extended our Discussion, and modified the overall language throughout the manuscript to reflect a broader discussion of the role of ghrelin signaling in the hippocampus and elsewhere in the brain in many aspects of feeding behavior. For example, our title now reads, "Hippocampus ghrelin signaling mediates appetite through lateral hypothalamic orexin pathways", and "learned feeding aspects of feeding behavior" was explicitly removed from the Abstract. Other instances include the Introduction and Discussion (end of last three paragraphs). While we retain the viewpoint that hippocampal ghrelin signaling is critical for conditioned feeding (based on Exp.1), in this revised manuscript, we do not disregard the possibility of the vHP-LHA circuitry as being involved in a variety of other food-motivated behaviors. Overall our extended discussion uses a more cautious tone with regards to vHP ghrelin having an exclusive role in conditioned feeding.

*If the authors want to broaden the role of ghrelin acting on the 'hippocampus to orexin neuron circuit' to mediate many aspects a feeding, then they should compare the relative importance of hippocampal input to orexin neurons with other potential inputs (reviewer #2 comment 3). The authors should also address the other concerns raised by the reviewers.*

As specifically suggested by Reviewer #2 we have added an additional experiment which compares the relative importance of hippocampal GHSR mediated conditioned feeding with another GHSR rich region, the lateral hypothalamus (LHA). As stated by Reviewer #2, the LHA contains orexin producing neurons that are activated by both LHA ghrelin infusion and by external food-related cues (Olszewski et al., 2003; Petrovich et al. 2012). Moreover, LHA ghrelin infusion (like vHP ghrelin infusion) increases food intake in rats (Olszewski et al., 2003). Therefore, the LHA is a logical additional target of potential importance for ghrelin-mediated conditioned feeding effects. Our new experiment replicates Exp.1, except that the GHSR-1A receptor antagonist, JMV2959, is now administered bilaterally to the LHA in meal-entrained animals and non-entrained animals (20 hr food deprived). We have included this new experiment in the manuscript, and the results (described in subsection “Endogenous ghrelin signaling in LHA neurons does not regulate conditioned feeding behavior”) are displayed in Figure 2—figure supplement 1. Our results show that GHSR blockade using a dose of JMV2959 that was effective in the vHP had no effect on entrained food intake or non-entrained food intake in similarly food-restricted animals. These findings suggest that vHP ghrelin signaling is particularly critical in the expression of conditioned feeding behavior based on internal (or potentially external) contextual cues, whereas LHA ghrelin signaling could be mediating other aspects of appetite and feeding. In this revised manuscript we now further elaborate in the Discussion on the feeding effects of ghrelin in brain regions such as the lateral hypothalamus, ventral tegmental area, and amygdala (Olszewski et al., 2003;, Perello et al., 2010; Skibicka et al., 2011; Alvarez-Crespo et al., 2012) as well as other feeding effects of downstream orexin signaling (Zhu et al., 2002; Goforth et al., 2014; Zheng et al., 2007; Parise et al., 2011; Harris et al., 2005; Thorpe et al., 2005; Thorpe et al., 2006). The specific mechanisms underlying LHA ghrelin mediated feeding will require further studies beyond the scope of this manuscript, but we believe that this additional experiment adds further support for the importance of vHP GHSR-1A signaling specifically in mediating conditioned feeding effects and improves the overall quality of the manuscript.

Minor comment from the Reviewing editor: It is customary to put a space between numbers and units, e.g. 4 hr, not 4 hr. Also, abbreviations should be used consistently, e.g. in the first paragraph of Results, we see 4 hr and 4h food intake. Finally, there is only one Fos gene in the rat, so there is no need to distinguish c-Fos from v-Fos (which is of viral origin); i.e., Fos is sufficient.

We thank you for the suggestion. Accordingly, we have now added spaces between numbers and units and we have corrected for consistency in abbreviations throughout the manuscript. We appreciate your suggestion regarding the ‘Fos’ abbreviation, but we have opted to stick with the ‘c-Fos’ abbreviation, as we are aware of reports indicating that the rat brain also expresses ΔFosB (Hollen et al., 1997).

Reviewer #1:

*[…] A major comment follows, although it should not require any major changes to address: The investigators frame this work and this pathway as all being relevant to ‘conditioned feeding’ which they model with about a week of restricted access to food for only 4 hr per day. However, I think it is premature with the data provide to attribute this pathway as being relevant specifically to their model of conditioned feeding. Only in the experiments summarized in Figure 2 was the meal entrainment done – and for those experiments, injections of a GHSR antagonist directly into the vHP were used to confirm that GHSR-expressing vHP neurons were important for conditioned feeding (but not for 4 hr rebound feeding following a 20 hr food restriction). In the subsequent experiments, neuroanatomic techniques were used to track the projections of GHSR-expressing/ghrelin-responsive neurons to the LHA. Also, the orexigenic effects of pharmacologic ghrelin-injections into the vHP were assessed in LHA-lesioned animals or Orexin receptor blocker-administered animals. Activation of LHA neurons was also assessed in animals administered ghrelin into the vHP. Thus, while GHSR-expressing vHP neurons appear to be important for conditioned feeding, the GHSR-expressing vHP neurons projecting to the LHA orexin neurons are not necessarily involved in that particular aspect of ghrelin-induced eating. Thus, the possible role of this newly described pathway in a broader scope of ghrelin's many feeding-related actions should be mentioned. If the investigators want to limit this pathway to conditioned feeding, further physiologic experiments would be needed – e.g., inhibit downstream portions of that pathway and determine how it affects conditioned feeding.*

We thank the reviewer for emphasizing an important caveat in our study; that vHP ghrelin to LHA neural circuitry could potentially mediate other aspects of feeding and appetite. We now take an overall more cautious approach regarding vHP ghrelin’s role in feeding as being exclusive to conditioned aspects. We have expanded our discussion of this novel vHP->LHA pathway to include the putative involvement of other (non-conditioned) aspects of feeding behavior. In addition, we have altered the title and Abstract to reflect this more cautious perspective. Please see above (response to Reviewing Editor) where we expand on how we address this caveat in this revision, and where we also explain the conceptual difficulties inherent in expanding Exps. 3 and 6 to include conditioned feeding models.

Reviewer #2:

*The set of studies presented here are important in addressing the neurobiological substrates of overeating in today's society – where consumption is rarely triggered by caloric deficit. They are also important in elucidating an under-appreciated node for the regulation of feeding – the vHP. The work is well conducted and thorough. The authors provide ample data to support most of their conclusions. I have three major comments that need to be addressed: 1) A key argument, and the title of the paper, posits that ghrelin action in the ventral hippocampus augments the conditioned aspects of feeding through LH orexin neurons. A limitation of the manuscript is that the authors do not actually show this. First, they show that hippocampal ghrelin augments conditioned bouts of feeding. They then go on to show that hippocampal ghrelin increases unconditioned food intake and this requires an intact ipsilateral LH (Figure 5) and the availability of central orexin receptors (Figure 8). However, the authors do not show that central orexin receptors are required for ventral hippocampal ghrelin to augment conditioned feeding. Repeating the approach in Figure 1 but infusing either vehicle or the SB334867 into the lateral ventricle would solidify the argument. The use of an orexin saporin for a selective LH lesion would also solidify the argument.*

We appreciate the reviewer for highlighting this limitation with the manuscript and for providing suggestions to enhance the paper. Please see above (response to the Reviewing Editor) for our rationale regarding the conceptual difficulty involved with repeating the experiments from Figures 5 and 8 using a conditioned feeding approach. Thus, as an alternative to address this caveat, we have now stressed throughout this revised manuscript that while vHP ghrelin signaling has important roles in conditioned feeding, it may well be the case that vHP ghrelin to LHA orexin neural circuitry has broader roles in feeding behavior, particularly given that orexin signaling and the LHA neural processing are involved in several aspects of feeding behavior (Zhu et al., 2002; Goforth et al., 2014; Zheng et al., 2007; Parise et al., 2011; Harris et al., 2005; Thorpe et al., 2005; Thorpe et al., 2006), (Stanley et al., 2011; Jennings et al., 2013; Legrand et al., 2015; Cason and Aston-Jones, 2013). Overall, in this revision we now take a more cautious approach towards attributing this circuit with an exclusive role in conditioned feeding, and we have altered the title, Abstract, and Discussion accordingly.

Regarding your suggestion to repeat the approach in Figure 1 with LV SB334867, this is an interesting idea but we do not think that such an approach is feasible as it would require an orexigenic agonist (vHP ghrelin) application under conditions of excessive feeding. In other words, in our meal entrainment schedule, the rats are robustly hyperphagic during the 4 hr food access period, and thereby activation of vHP ghrelin receptors would be unlikely to augment intake from a high baseline of excessive food intake (i.e., ceiling effect).

*2) In Figure 5, the authors compare food intake induced by vHP ghrelin (or vehicle) in animals with (or without) LH lesions. They compare these groups separately according to whether or not the animals had an LH lesion or sham (Figure 5). These data should be combined and compared using an ANOVA with main effects of surgery (Lesion, Sham) and treatment (Vehicle, Ghrelin). This comparison will help parse out any non-specific effects of the LH lesion unrelated to ghrelin treatment. Moreover, this is critical because bilateral LH lesions are well known to alter feeding behavior and it appears there are differences in the amount of food consumed between the lesion and sham animals that received vehicle. For example, compare food intake vehicle treated animals in lesion and sham groups at 1 hr and 5 hr.*

We thank the reviewer for bringing our attention to this important matter. It was an oversight to omit this important statistical analysis from our previous submission, and we have now included the analysis in accordance with the reviewers; suggestion. After performing an ANOVA with group as a between subjects factor and drug as a repeated measures factor, we found no overall significant group effect on food intake at any time point. Importantly, we did obtain a significant interaction between group and drug at the 5 hr time point of food intake, and marginal significance for this interaction at 1 hr and 3 hr. This analysis is now added to the results of Exp.3 (subsection “Endogenous ghrelin signaling in LHA neurons does not regulate conditioned feeding behavior”).

*3) The authors are very thoughtful in their Discussion. However, if the authors posit that the source for vHP GHSR activation is peripheral ghrelin evoked by interoceptive or external cues then GHSRs are likely to be activated in other brain regions where they are expressed – including LHA GHSRs. LHA orexin neurons express GHSRs and are activated by cues. While the authors are to be commended for showing that vHP GHSRs are required for the full expression of conditioned feeding in the present paradigm – supporting an endogenous roll for ghrelin in the vHP, the manuscript would be strengthened by investigating another GHSR-rich region to determine if similar effects are observed or whether the GHSRs in the vHP uniquely participated in meal-entrained conditioned feeding. At the very least, this unresolved issue should be discussed.*

We thank the reviewer for this suggestion. We agree that ghrelin signaling in different brain regions likely produces diverse effects on feeding behavior, which has prompted us to perform your suggested experiment: examining another GHSR-rich region (the LHA) and its function in conditioned feeding. We administered the ghrelin receptor antagonist JMV2959 in the LHA (using a dose that was effective in the vHP for reducing meal-entrained intake) in both meal-entrained and non-entrained (20 hr restricted) animals. Our results from this new experiment are displayed in Figure 2—figure supplement 1 (please also see “Experiment 1: Effect of vHP-mediated ghrelin signaling on conditioned feeding behavior”, in the Methods, and “Endogenous ghrelin signaling in LHA neurons does not regulate conditioned feeding behavior”, in the Results). We found that LHA GHSR blockade had no effects on food intake in meal entrained and non-meal entrained animals under the same deprivation conditions, using an identical design as experiment 1. These data suggest that ghrelin signaling in the LHA is not involved in conditioned feeding, (or at least to a lesser extent than the vHP) and may potentially have a broader and/or fundamentally different role in feeding behavior. Further assessment of LHA ghrelin mediated feeding behavior is beyond the scope of this manuscript and will require future experiments. Importantly, the notion that vHP ghrelin signaling is uniquely involved in conditioned feeding remains supported by the new data set, yet we still take a more cautious tone regarding this perspective in the revised manuscript.

Following the reviewers’ suggestion, we have also expanded the Discussion to now address the role of ghrelin signaling in different brain regions as being involved different aspects of feeding (Sibicka et al., 2011; Sibicka et al., 2013). We now highlight two studies in which ICV (Salome et al., 2009) or intra-amygdala administration of JMV2959 (Alvarez-Crespo et al., 2012) reduces feeding in fasted (non-entrained) rats. Taken together, our study adds a novel perspective of ghrelin signaling in the brain by highlighting dissociative feeding effects of ghrelin depending on brain region.

References:

Perello M, Sakata I, Birnbaum S, Chuang JC, Osborne-Lawrence S, Rovinsky SA, et al. (2010): Ghrelin increases the rewarding value of high-fat diet in an orexin-dependent manner. *Biological psychiatry*. 67:880-886.

Hollen KM, Nakabeppu Y, Davies SW (1997): Changes in expression of delta FosB and the Fos family proteins following NMDA receptor activation in the rat striatum. Brain research Molecular brain research. 47:31-43.

Jennings JH, Rizzi G, Stamatakis AM, Ung RL, Stuber GD (2013): The inhibitory circuit architecture of the lateral hypothalamus orchestrates feeding. Science. 341:1517-1521.

Legrand R, Lucas N, Breton J, Dechelotte P, Fetissov SO (2015): Dopamine release in the lateral hypothalamus is stimulated by alpha-MSH in both the anticipatory and consummatory phases of feeding. Psychoneuroendocrinology. 56:79-87.